prolonged grief disorder; global; bereavement; grief reactions

**Corresponding author:**
Charlotte Elize Hilberdink;
Email: hilberdink@cyceron.fr

# Bereavement issues and prolonged grief disorder: A global perspective

Charlotte E. Hilberdink[1] , Kevin Ghainder[2], Alexandre Dubanchet[2], Devon Hinton[3,4], A. A. A. Manik J. Djelantik[5], Brian J. Hall[6] and Eric Bui[1,2,3]

[1]Normandie Univ, UNICAEN, INSERM, U1237, PhIND "Physiopathology and Imaging of Neurological Disorders", NEUROPRESAGE Team, (Institut Blood and Brain @ Caen-Normandie), GIP Cyceron, Caen, France; [2]Centre Hospitalier Universitaire Caen Normandie, Caen, France; [3]Massachusetts General Hospital, Boston, MA, USA; [4]Harvard Medical School, Boston, MA, USA; [5]Utrecht Medical Center, Utrecht, The Netherlands and [6]Center for Global Health Equity, New York University, Shanghai, 200122 People's Republic of China

## Abstract

The death of a loved one – bereavement – is a universal experience that marks the human mental health condition. Grief – the cognitive, emotional, and behavioral responses to bereavement – is thus experienced by virtually everyone at some point in life, while mourning is a process through which grievers come to terms with the loss envisioning life without the deceased. Although distress subsides over time among most bereaved individuals, a minority will develop a condition recently identified as prolonged grief disorder (PGD). The present review provides a global perspective on bereavement, grief reactions, and PGD. Although the loss of a loved one and grief reactions are in general experienced consistently across different cultures, differences and variations in their expression may exist across cultures. Especially within specific populations that may be more at risk for PGD, possibly due to risk factors associated with the mechanisms of loss (e.g., refugees, migrants, and conflict survivors). The diagnostic criteria for PGD are mostly based on Western grieving populations, and cultural adaptations of PGD treatments are limited. Therefore, cross-cultural development and validation of PGD screening/assessment is critical to support future research on grief reactions and PGD, especially in non-Western contexts, and concerning the potential future global changes and challenges that appear to have a major impact on PGD. More transcultural research on PGD is needed to contextualize and will lead to culture-bound symptom identification of PGD, and the adaptation of current treatment protocols, which may ultimately improve health at the individual level, and health-care systems.

## Impact statement

A minority of people who experience cognitive, emotional, and behavioral grief responses to bereavement will develop a mental health condition recently identified as prolonged grief disorder (PGD). In this review, we provided a global perspective on bereavement, grief reactions, and PGD, and we suggested that grief reactions seem to be consistent across different cultures, although differences and variations in the expression of symptoms may exist across cultures. Given the heterogeneity in PGD prevalence estimates across the globe, specific populations may be at more risk for PGD, possibly due to risk factors associated with the mechanisms of loss, such as population specificities (e.g., refugees, migrants, and conflict survivors).

As evidence-based treatment for PGD has limited cultural adaptations, cross-cultural development and validation of PGD screening and assessment may be critical to support future research on grief reactions and PGD. We therefore suggested that more transcultural research is needed to contextualize PGD, which may lead to the identification of culture-bound symptoms of PGD and the adaptation of current treatment protocols. Thereby it could improve health at the individual level and health-care systems for people who suffer from PGD. In general, there is a lack of trained therapists worldwide and existing PGD therapy is costly in therapist time. We will therefore need more cost- and time-efficient health-care strategies if we want to be able to address this major public health problem, given the sheer number of individuals suffering from PGD worldwide. To be able to address PGD at the global level, we will thus need to develop a stepped-care model in which the most effective, yet least resource-taxing treatment is delivered first, only *stepping up* to more resource-using treatments as required, which could decrease the health burden at the individual's level while controlling costs and improving health-care systems efficiency.





## Grief reactions across cultures

### From acute grief to integrated grief or prolonged grief

The death of a loved one is one of the most painful experiences in one's life. Most bereaved individuals experience distressing grief reactions to such a loss, including intense longing for the deceased, emotional pain and/or pangs of grief, loss of interest in ongoing life, and social withdrawal (Bui, 2018). These 'acute grief' responses are normal reactions to the death of a loved one that occur during the period that immediately follows the loss (Shear, 2012). This acute phase usually subsides in the first few months following the loss, and most individuals will successfully adjust, regaining their interest and engagement in their ongoing life (Bonanno and Kaltman, 1999; Shear et al., 2011). Transitioning from this state of 'acute grief' to a state of 'integrated grief', in which the grief reactions and distress are still present but no longer pervasive nor impairing, requires acknowledgment of the reality of the loss, integration of positive and negative emotions and cognitions regarding this loss, and successful updating of the working model (i.e., envisioning the world and the future without the deceased [Shear and Shair, 2005; Shear and Mulhare, 2008]). Yet, a substantial minority of bereaved individuals do not successfully adapt, in which grief reactions become pathological and symptomatic, and they exhibit continued, prolonged, distressing, and impairing grief reactions that can last years and sometimes decades after the death (Shear et al., 2013; Lundorff et al., 2017; Djelantik et al., 2020). For example, according to data from three bereavement and grief studies worldwide, 5–15% of community-based bereaved had a diagnosis of PGD one year after losing a loved one (Prigerson et al., 2021a).

### Prolonged grief disorder in ICD-11 and DSM-5

After decades of heated debate about the inclusion of a psychiatric mental health condition characterized by prolonged, distressing, and impairing grief into international nosological classifications (Lichtenthal et al., 2004; Shear et al., 2011; Bryant, 2012), prolonged grief disorder was included in the 11th Edition of the International Classification of Diseases in 2018 (ICD-11, World Health Organization [WHO], 2022). In 2022, PGD was also added to the text revision of the Fifth Edition of the Diagnostic and Statistical Manual of Mental Disorders (DSM-5-TR, American Psychological Association [APA], 2022) under the trauma and stressor-related disorders section. Although the classifications for PGD are included both in the ICD-11 and the DSM-5-TR, they only show moderate overlap as they differ in their number and content of symptoms, time criteria (6 months for ICD-11 and 12 months for DSM-5-TR), and diagnostic algorithms (Boelen and Lenferink, 2020; Prigerson et al., 2021b). A detailed overview of these discrepancies is given by Eisma (2023). An overview of the diagnostic criteria for the ICD-11 and DSM-5-TR is provided in Table 1. For the rest of this review, we will use the term PGD to refer to this clinical condition of maladaptive prolonged grieving, although previous research has used other terms including complicated grief, or traumatic grief.

### Grief reactions across cultures

Culture is defined as a set of traditions, rituals, values, and beliefs that are shared between members of a group of human beings (Jones, 2005). Although bereavement is a life stressor experienced ubiquitously across the globe, some data suggest that normal grief reactions might vary in their expected expression, their length, and their intensity across cultural and religious backgrounds (Eisenbruch, 1984). This might be the case because of the way people across different cultures communicate and express their grief responses (i.e., which pathological reactions they report), the existence of culture-bound symptoms, variations in help-seeking behavior, type of help and access to social support, culture-specific coping strategies, as well as variance in the meanings of symptoms and disorders, and stigmatization concerning mental health problems (Gureje et al., 2019, 2020). Therefore, cultural rituals, practices, and societal norms might have a significant impact on the way people grieve, mourn, and adjust to a loss (Casarett et al., 2001; Rosenblatt, 2001). The cultural norm hypothesis offers a potential explanation for cultural variability in the expression and duration of pathological grief, as this hypothesis suggests that disrupted emotional behavior, such as emotional reactivity, will become dysregulated and symptomatic if they differ from their culturally-determined normative or standard experiences and expressions (Chentsova-Dutton et al., 2010). For example, European Americans with depression, who normally express their emotions openly, showed diminished positive emotional reactivity to an amusing movie clip than those without depression; however, Asian Americans with depression showed similar and increased emotional reactivity to the same clip, although they normally dampen and control their emotions. Grief expressions and duration may thus be dependent on cultural norms and beliefs about emotional behavior, and could complicate cross-cultural interpretation of symptoms and behaviors, and influence diagnostic validity and accuracy.

### Variations across cultures in the expected normal expression of grief

Although there are communalities in the (non)pathological responses of grief across cultures, as, for example, all (Western) PGD symptoms were recognized as familiar grieving and mourning emotions in the Balinese society (Djelantik et al., 2021), the expression of normal grief reactions and PGD symptoms may also be culturally sensitive, varying across societies and ethnic populations. For example, a study on Cambodian refugees reported that dreams and hallucinations of the deceased are a crucial part of normal grief and bereavement, whereas these types of re-experiences are often considered pathological in Western cultures (Hinton et al., 2013). Also, the Chinese bereaved may report more somatic stress symptoms in response to bereavement, including increased headaches and stomach or back pains (Ho et al., 2002). In line with this, Swiss and Chinese parents also exhibit differences regarding their symptom patterns after the loss of a child at least 6 months ago; Swiss ones experienced more severe grief-related preoccupation, whereas their Chinese counterparts reported more functional impairments, depression, and a sense of a meaningless and empty life (Xiu et al., 2016). Further, a similar expression may be defined as pathological and symptomatic in some cultures, whereas they may be considered part of the normal mourning process in other cultures (Li and Prigerson, 2016). Finally, the actual expression of grief reactions across cultures seems to be influenced by cultural norms and beliefs. For example, widows in Taiwan are highly encouraged not to cry in front of the deceased (Hsu et al., 2004).

### Variations across cultures in the expected normal duration of grief

There also appears to be variation across cultures in the expected length of mourning, during which 'acute grief' is considered

**Table 1.** The diagnostic criteria of prolonged grief disorder (PGD) according to the 11th Edition of the International Classification of Diseases (ICD-11) and the text revision of the Fifth Edition of the Diagnostic and Statistical Manual of Mental Disorders (DSM-5-TR)

| Diagnostic criteria ICD-11 | Requirements for diagnosis |
|---|---|
| **Criteria A** Exposure | A history of bereavement following the death of someone close |
| **Criteria B** Types of grief reactions | *If the following persistent and pervasive grief reactions are present:* • Longing for the deceased • Persistent preoccupation with the deceased |
| **Criteria C** Accompanying impairments | *If the following impairments are present:* • Intense emotional pain (e.g., sadness, guilt, anger, denial, and blame) • Difficulty accepting the death • Feeling that one's has lost a part of one's self • An inability to experience a positive mood • Emotional numbness • Difficulty engaging with social or other activities |
| **Criteria D** Significant distress/impairment | If the impairments of criteria B and C result in significant impairment in daily life (e.g., personal, family, social, educational, occupational, or other areas of functioning), or if functioning is maintained, but only through significant additional effort. |
| **Criteria E** Cultural and time | If the pervasive grief reactions and impairments of criteria B and C have persisted for an atypically long period of time after the loss, markedly exceeding expected social, cultural, or religious norms for the individual's culture and context, except if grief responses last for <6 months, of which this duration norm can depend on some cultural contexts. |
| **Diagnostic criteria for DSM-5-TR** | **Requirements for diagnosis** |
| **Exposure** | Bereavement – the death of someone close |
| **Criteria A** Timing | Loss experienced… >12 months prior in adults >6 months prior in children/adolescents |
| **Criteria B** Types of grief reactions | *If the following grief reactions are present, nearly daily in the prior month:* • Intense yearning and longing for the deceased • Preoccupation with thoughts or memories of the deceased |
| **Criteria C** Accompanying impairments | *If at least three of the following impairments are present, nearly daily in the prior month:* • Marked sense of disbelief about the loss • Identity issues (e.g., feelings as if a part of oneself has passed) • Avoidance of reminders of the deceased • Emotional pain • Decreased emotional experiences and numbness • Meaninglessness • Intense loneliness • Difficulties in social interactions and/or engaging in daily activities due to the loss |
| **Criteria D** Significant distress/impairment | If the impairments of criteria B and C cause significant distress and/or impairment. |
| **Criteria E** Cultural and contextual norms | If the duration and severity of the impairments of criteria B and C exceed the individual's expected social, cultural, or religious norms. |
| **Criteria F** Other mental health conditions | Grief reactions and impairments differentiate from the physiological effects of substance use, a major depressive disorder, posttraumatic stress disorder, or any other mental or medical disorder. |

normal. For example, in Germany, a Western society in which a Christian tradition prevails, a mourning period of one year is culturally accepted (Hays and Hendrix, 2008). This mourning period can be much longer in non-Western cultures, such as traditional Chinese culture. In some Chinese traditions, a long-lasting self-restricted period of three years is expected after the loss of a child or parent, which implies respect for the deceased relative (Braun and Nichols, 1997; Klass and Chow, 2011). In Bali, Indonesia, numerous mourning rituals with social gatherings take place up to 10 years after the loss (Djelantik et al., 2021). This may indicate a cross-cultural caveat in regard to the expected duration of a 'normal' mourning process (Rosenblatt, 2008), with implications for the validity of PGD diagnostic criteria across cultures.

## Variations across cultures in mourning rituals

Although most religion-related rituals have been declining in Western cultures, they seem to show a large cross-cultural overlap. Rituals mostly focus on the transfer of the deceased to another life or a ruler of a religious belief (Lobar et al., 2006), and provide a communal framework to increase an individual's ability to adjust to changes in social identifies, (re)constructing social relationships, and form a memory of the deceased within the society (Silverman et al., 2021). Rituals also assist in later apology from the bereaved to the lost one and gratitude for the deceased (Hinton et al., 2013), and therefore offer a context to appropriately and safely express strong emotions by focusing on meaning-making, social narratives, as well as cultural standards. For example, a 2-day mourning ritual of the

Papua New Guinean society, Lihir, consists of an initial emotional phase followed by a phase of forgetting and remembering (Hemer, 2010). Also, a funeral provides a safe and structured event to make space for the mourning process (Giblin and Hug, 2006; Mathew, 2021). On the other hand, Western grief rituals seem to mostly concentrate on individual grief and include less social context compared to most non-Western practices. For example, in Balinese mourning rituals, the purpose is predominantly focused on expressing care and respect for the deceased, and not on an individual's own emotional grief process. It could be that this serves as a protective cultural practice against the development of mental health issues (Djelantik et al., 2021). A recent review of 22 studies on ritual and symbolic interventions as sizeable part of grief therapy accordingly showed significant positive effects on trauma-related symptoms in individuals with PGD, although these effects were mostly an effect of the treatment as a whole and not depending on specific ritual interventional elements (e.g., symbolic expression, farewell ceremonies, dialogue with the deceased, [Wojtkowiak et al., 2021]).

All these mourning rituals have thus been suggested to prevent the development of PGD, by providing specific guidance for grief recovery, offering coping strategies, and direct the process of the death (Cacciatore and DeFrain, 2015), as well as giving social support during the mourning period. However, their protective effect may be compounded by confounding factors such as spiritual and religious beliefs and affiliations that may be associated with the development of PGD, though the direction of this association varies across study settings and populations (Easterling et al., 2000; Hebert et al., 2007; Schaal et al., 2014; Christian et al., 2019). Here again, the cultural context might moderate the relationship between mourning rituals (and spirituality in a global sense) and the development of PGD.

### Prolonged grief disorder prevalence across the world

Research has investigated the prevalence of PGD across the globe using mostly standard grief questionnaires, developed in and for Westernized countries (North American and Europe). Below we describe these studies.

### All deaths

According to the meta-analysis of Lundorff et al. (2017), a pooled prevalence rate of 9.8% (95% CI 6.8–14.0) was found in 14 studies across the world (pooled N = 8,035) among bereaved adults who experienced mostly nonviolent deaths of a loved one. In Westernized countries, which consist of higher-income countries based on the Organization for Economic Co-Operations and Development (OECD, 2023), studies have reported somewhat heterogeneous but relatively high prevalence rates for PGD across populations (children, adults) and across types of deaths. For example, cross-sectional studies reported prevalence estimates among bereaved adults who lost their partner, friend, family member, or other loved one due to mostly natural deaths ranging from a lower rate of 2% in Israel (Killikelly et al., 2019) to a high rate of 25.4% in the Netherlands (Newson et al., 2011), although the latter included somewhat older bereaved adults. This is comparable to prevalence among bereaved college students (aged >18 years old) from the United States (US), ranging from 1.7% (Balk et al., 2010) and 13.4% (Glickman, 2020) in an ethnically diverse student population. Although few studies exist on PGD prevalence in younger populations, the

few studies that estimated the prevalence of PGD among bereaved children and adolescents were comparable, ranging between 1.2–6.7% in a German population including adolescents and adults (Kersting et al., 2011; Rosner et al., 2021) and 3.4–12.4% in Dutch children (Boelen et al., 2019).

In non-Westernized lower-to-middle-income countries, which predominantly consist of most Asian countries, Latin America, and the Middle East and North Africa (OECD, 2023), PGD prevalence appears to be relatively higher than in high-income countries, but also shows heterogeneity across countries. Prevalence of PGD after loss mostly due to a nonviolent natural death was found to be between 7.1–12.6% in Chinese adults (Killikelly et al., 2021), but lower (pooled 3.7%, 95% CI 3.1–4.5%, N = 2,524) in an African adult population (Ghana: 2.6%, Kenya: 3.4%, Nigeria: 4.6% [Ben-Ezra et al., 2020]). Cross-sectional studies that studied prevalence in bereaved adolescents as well as adults, found a rate of 1.8% in Chinese individuals (He et al., 2014) and 12% in Saudi Arabian university students (Al-Gamal et al., 2019), who were adolescent or adult.

### Violent deaths

Higher prevalence estimates have been found in bereaved after losing a loved one due to an unnatural cause of death. Djelantik et al. (2020) found a pooled PGD prevalence of 49% (95% CI 33.6–65.4%) in N = 4,774 adults across the globe who experienced an unexpected and/or violent death (traumatic accidents, suicide, homicide, or disasters) in 25 studies across OECD and non-OECD membership countries. Studies in bereaved individuals due to unnatural deaths predominantly investigated PGD after the (unexpected) loss of a child, the loss of a family member, close friend, or other relative in bereaved caregivers, or survivors of a (natural) disaster. PGD prevalence estimates among bereaved caregivers seem to vary greatly according to studies across Westernized countries. For instance, Swiss adult caregivers had a relatively lower prevalence of 4.9% (Wagner et al., 2012), whereas in Portuguese caregivers a higher prevalence estimate of 28.8% was reported 6 months post-loss (Coelho et al., 2015), although samples were not necessarily representative and therefore difficult to compare. In bereaved adults who lost a close family member, friend, or colleague 2.5–3.5 years after the 9/11 terrorist attacks in New York, the US, even a higher prevalence of 43% was reported (Neria et al., 2007).

The existing cross-sectional studies on PGD in non-Westernized populations predominantly investigated adult refugee populations living in Westernized countries, as they are at high risk for traumatization and loss, and relatively high prevalence rates have been reported. For example, bereaved female refugees from several Middle East and Northern African countries, who were living in shared accommodations in Germany at the time PGD symptoms were measured, had a somewhat lower PGD prevalence of 9.4% (Steil et al., 2019), although also high rates of 54% have been reported in Bosnian refugees who were living in the southeastern US for more than 5 years (Craig et al., 2009).

Prevalence estimates in non-Westernized populations living in non-Western countries appear to be somewhat lower than in refugee populations, although still relatively high; in adult survivors of the Rwandan genocide, PGD prevalence was 8% after losing a family member, child, or parent (Schaal et al., 2010), and was found to be 14.3% in Cambodian adult survivors of the Khmer Rouge regime after losing at least one family member due to a war-related violent death (Stammel et al., 2012). A recent meta-analysis accordingly showed that PGD prevalence rates in populations living in

higher-income Westernized countries were lower than in lower-income non-Western countries (Djelantik et al., 2020). Furthermore, after the loss of a child, spouse, parent, or other family member due to a disaster, a prevalence of 8.5% was reported in bereaved adult survivors of the Wenchuan earthquake in China (Yi et al., 2018). The limited studies that exist on children and adolescents were mostly included in a pooled population that also included adults. One study, for example, found a PGD prevalence of 15.8% after losing someone close in a bereaved refugee population consisting of children, adolescents, and adults who were in Australia (Bryant et al., 2019). Moreover, a study on child, adolescent, and adult survivors of a tsunami in India reported that 25.9% had PGD after losing a spouse, child, or family member (Rajkumar et al., 2015), although this was found to be 34.6% in bereaved children after losing a parent due to war-related violence in Kosovo (Morina et al., 2011). In contrast, in a study among 301 bereaved family members of traffic accidents in Bali, Indonesia, PGD symptoms existed but none met the criteria for PGD (Djelantik et al., 2021).

## Risk factors for prolonged grief disorder in the global context

Previous studies identified several factors present before, during, and after the loss, or related to the loss that increased or decreased the risk for PGD development. However, there are indications of important cross-cultural discrepancies between Westernized and non-Westernized countries that define the specific risk for PGD, as some factors may not be applicable across different cultures (Chukwuorji et al., 2018) or could be attributable to racial and socioeconomic inequalities. These risk factors exist globally, but may be more pronounced in the non-Western World.

### Pre-loss risk factors

Studies found several demographic factors that increase the risk for development of PGD, such as lower socioeconomic status, being female, and being older (Gilbar and Ben-Zur, 2002; Ferrario et al., 2004; Simon et al., 2005; Vanderwerker et al., 2006; Kersting et al., 2011; Newson et al., 2011; Sung et al., 2011; Hu et al., 2015; Fernández-Alcántara and Zech, 2017; Heeke et al., 2017; Lundorff et al., 2017; Specht et al., 2022). As these factors vary across cultural contexts and are related to cultural diversity, such as race, common mental health problems, socioeconomic constructs, and cultural norms and beliefs, they might affect the experience of losing a loved one and consequently the likelihood of developing PGD. For example, a study that compared Westernized French and (non-Westernized) Togolese bereaved populations found that being male was a significant predictor for PGD in Togolese, but not in French adults (Kokou-Kpolou et al., 2020). Other personal characteristics present before the loss, such as exposure to childhood adversity, psychological and physical health history, pre-loss alcohol consumption, and maternal age in case of losing an infant, appear to influence PGD risk (Simon et al., 2005; Sung et al., 2011; Goldstein et al., 2019). As some pre-loss factors may be more common in specific cultures and the perception of certain factors appears to be culture dependent, they might vary in their risk for PGD. For example, in a population that mostly consisted of South African women, having depressive symptoms, older maternal age, and consuming a minimum of two alcohol units per day before the loss predicted PGD diagnosis during the two years after losing their infant (Goldstein et al., 2019).

### Loss-related risk factors

Studies also found factors that are related to the loss and increased risk for PGD, such as the type and duration of the relationship with the deceased (Kokou-Kpolou et al., 2020; Smith and Ehlers, 2020). For example, Burke and Neimeyer (2012) reported in their review that PGD is more severe in individuals that have lost a close family member, spouse, or life partner compared to those who have lost a distant relative. Similarly, losing or missing a close family member and uncertainty about knowing who does or does not belong to the family system significantly predicted prolonged grief severity in treatment-seeking Syrian refugees (Renner et al., 2021). The number of family members additionally appears to be a significant predictor for PGD, as more core family members lost was associated with PGD severity in asylum seekers living in collective accommodations in Germany (Comtesse and Rosner, 2019). Also, elderly who had lost a spouse were at increased risk for developing grief symptoms than those who lost someone else (Fujisawa et al., 2010). Furthermore, as discussed above, an increased prevalence of PGD has been reported in the case of unexpected and/or violent death (traumatic accidents, suicide, homicide, or disasters [Djelantik et al., 2020]). The subjective impact of a life stressor (e.g., life-threatening situation, death of a loved one) may be indexed by psychological reactions occurring during or immediately after exposure to a major stressor. Those peritraumatic reactions (distress and dissociation) have been shown to consistently predict the development of posttraumatic stress disorder across populations (e.g., Bui et al., 2010; Vance et al., 2018), and recent data suggest that they may also predict the development of PGD in adults (Hargrave et al., 2012; Mutabaruka et al., 2012; Bui et al., 2013) and children (Revet et al., 2021).

### Post-loss risk factors

The lack of social support after a loss has been identified as a risk factor for the development of PGD (Stroebe and Schut, 2001; van der Houwen et al., 2010). Culture could be a determining factor in social support, as for example more individualistic societies were hypothesized to isolate themselves more and have more difficulties with receiving and asking for professional support (Kumar et al., 2012). Yet, some cultural norms could also be of importance for this, as in Asian populations, it is less common to ask for and accept social support for coping with stress, which could increase PGD risk (Hashimoto et al., 1999; Taylor et al., 2004; Taylor and Lynch, 2004; Kim et al., 2008; Shi et al., 2020). Finally, low social support may be a risk factor for PGD (Vanderwerk and Prigerson, 2004) and may vary across cultures (Hofstede and Hofstede, 2005; Bauman, 2012; Kumar et al., 2012).

## Discussion

The present paper reviewed grief reactions and symptoms across cultures, the prevalence estimates, and risk factors for PGD across countries and regions of the globe that may impact PGD on the global stage. Implications of our findings include recommendations for researching, assessing, preventing, and treating PGD across the globe.

Prevalence estimates for PGD appear to be quite similar across Western and non-Western countries (e.g., Killikelly et al., 2018), with the apparent higher rates in non-Western countries (at least partially) explained by confounding factors including the specificities of the specific populations studied (e.g., refugees, migrants, and conflict survivors) that may be more at risk for PGD (Nickerson et al., 2014; Hollander, 2016; Chukwuorji et al., 2018; Comtesse et al., 2021). Overall, the existing literature supports that about 10% of individuals have been shown to go on developing PGD after the loss of a loved one (Lundorff et al., 2017). Yet, differences in prevalence rates worldwide could also be related to different diagnostic criteria sets and tools, and there is a lack of culturally-specific research on these criteria (Stelzer et al., 2020). Therefore, no globally applicable criteria set exists yet.

So far, most measures used to assess PGD have been developed in the US and their validity is yet to be established across cultures. For example, the Inventory of Complicated Grief, the measure that has been most used to assess pathological grief, has been validated in eight languages, including Hebrew (Lifshitz et al., 2020), Korean (Han et al., 2016), Polish (Ludwikowska-Świeboda and Lachowska, 2019), Italian (Carmassi et al., 2014), and Spanish (Spanish version [Limonero et al., 2009], Colombian-Spanish version [Gamba-Collazos et al., 2017], Mexican-Spanish version [Dominguez-Rodriguez et al., 2021]). However, it is based on older diagnostic criteria for PGD. More recently developed measures of PGD, including the prolonged grief disorder scale (PG-13) and Structured Clinical Interview for Complicated Grief (SCI-CG), do assess the ICD-11 and DSM-5-TR diagnostic criteria, but are only available in a limited number of languages (Bui et al., 2015; Prigerson et al., 2021a). Is it however important to keep in mind that currently available data on PGD symptom severity and prevalence rates is based on both (self-reported) diagnostic questionnaires and clinical diagnostic interviews, which could hamper the generalizability of study results. Validating and/or developing screening and assessment measures that are reliable across cultures is critical to support future cross-cultural research on grief reactions and PGD. The International Prolonged Grief Disorder Scale assesses PGD for the ICD-11 and includes a cultural supplement that offers information on culturally-relevant grief symptoms (Killikelly et al., 2020). This tool seems to be suitable for operationalizing the cultural caveat for PGD in specific clinical settings, and thus might improve global applicability for researchers and clinicians, although further research is needed (Killikelly and Maercker, 2022).

Further, although certain types of grief reactions and symptoms have been consistently found across different cultures, including difficulties accepting the loss, losing interest in relationships, and feeling dazed and stunned (Xiu et al., 2016; McNeil et al., 2020), differences and variations in symptom expression may exist across cultures. In fact, diagnostic criteria for PGD were mostly based on knowledge of Western grieving populations (Prigerson et al., 2009; Boelen et al., 2010; Newson et al., 2011; Schaal et al., 2014), and though yearning or longing for the deceased may be found across cultures (Prigerson et al., 2009; He et al., 2014; Li and Prigerson, 2016; Xiu et al., 2016), there may be other culturally-specific symptoms of grief that are not currently included in PGD criteria sets (Killikelly et al., 2018). Thus, important cross-cultural knowledge is still lacking and hampers the validity of DSM-5-TR or ICD-11 PGD diagnostic criteria across other cultures, and further research on the specific cultural expressions and symptoms of PGD, and how these differential symptom expressions may impact screening and treatment strategies, are warranted.

To date, the treatment with the most empirical support for PGD is a 16-session individual therapy that has shown efficacy across three large randomized controlled trials (Shear et al., 2005; 2014; 2016). However, the three trials have been conducted in the US, limiting the generalizability of the results to populations outside the US and outside the Western World. The identification of key culture-bound symptoms of anxiety and trauma-related disorders has previously supported their inclusion in DSM-5 and shows how trauma may result in particular symptoms in a specific cultural context (Hinton et al., 2010), and the development of specific treatment protocols (Hinton et al., 2005). It is thus possible that more transcultural research on other trauma-related issues such as PGD will lead to the identification of specific culture-bound symptoms of PGD and the adaptation of current treatment protocols.

Next to the differences and variations in PGD between Western and non-Western cultures, which seem to be mostly confounded by exposure to unsafe, violent, and unpredictable living areas, this diversity may certainly also exist within Western or non-Western cultures across different living areas. This also applies to diversity in religions and beliefs across the entire globe, even within specific countries, and to different norms in general that exist across and within cultural contexts which can be individually determined within a culture or within a specific cultural subgroup. It seems that the main dimensions of cross-cultural differences relevant for PGD across the globe are individualism, social(economic) support, coping mechanisms, and accessibility to mental health systems. These findings are not solely important for PGD, but might additionally offer important implications for cross-cultural differences for other trauma- and nontrauma-related health problems.

Recent changes on the global stage could also be of importance to acknowledge as they also might have a major impact on PGD across populations, countries, and cultures. Situations that create a risk for PGD development all have in common the occurrence of many sudden and/or violent deaths within a community in a limited timeframe. This could exceed the capacity to offer appropriate social support and could be associated with other risk factors such as ongoing stressors (e.g., material, fear of being contaminated, social distancing, and financial insecurity), such as during the COVID-19 pandemic and climate change. The COVID-19 health crisis, for example, could have been a risk factor for PGD and blocking the mourning process as the death was accompanied by extremely inconvenient circumstances (e.g., no possibility for funeral rites), feelings of guilt (potential contamination), and suddenness of the death. Although some studies demonstrated increased impact of COVID-related losses on PGD severity compared to natural losses (e.g., Eisma and Tamminga, 2020; Eisma et al., 2021; Edwards et al., 2023), other studies did not find a different impact of COVID- versus nonCOVID-related (unnatural, natural) deaths on PGD severity (Lenferink and Boelen, 2023). Differences between studies may be due to methodological differences including the used measurement tools and time since loss. The exact impact of the COVID-19 pandemic as a risk factor for PGD remains to be investigated. Moreover, climate change and potential associated disasters (e.g., hurricanes, floods, and droughts) may directly as well as indirectly contribute to an increased risk for developing PGD worldwide (Scott et al., 2012a, 2012b), as they include sudden and/or violent deaths, with financial and material insecurity, and disorganized social support systems. Taken together with recent trends worldwide towards individualism and social isolation (Hofstede and Hofstede, 2005; Bauman, 2012; Kumar et al., 2012), major world events such as pandemics, conflicts, and climate-change-induced natural disasters could have

a profound impact on rates of PGD in the world in the near future. This could be in line with the fact that prevalence estimates seem to be highest after a loss that is characterized by unexpectedness and with a violent and unnatural feature.

As we indicate, an evidence-based treatment for PGD is available (Shear and Gribbin Bloom, 2017), but with limited cultural adaptations. Further, there is a lack of trained therapists worldwide, and although this therapy is efficacious, it is costly in therapist time. About 16 h of a trained therapist is needed to successfully treat an individual with PGD, and given the sheer number of individuals suffering from PGD in the world, we will need more cost- and time-efficient health-care strategies if we want to be able to address this major public health problem. Strategies to address mental (and physical) health problems after a major life stressor such as bereavement exist on a continuum from psychoeducation to relaxation techniques, to medications and psychotherapy with a trained professional. To be able to address PGD at the global level, we will need to develop a stepped-care model in which the most effective yet least resource-taxing treatment is delivered first, only *stepping up* to more resource-using treatments as required; this could decrease the health burden at the individual level while controlling costs and improving health-care systems efficiency. For instance, integrating a phased approach and looking into potential models could guide the allocation of resources that are needed at the efficient level (system level, self-directed level, or mental health-supported brief interventions) and timing (during the initial impact, post-impact, or longer-term after the impact) and might ultimately provide evidence-based clinical care (Rauch et al., 2021). To facilitate more cross-cultural research on specific cultural aspects, influences, and cultural-bound risk factors for PGD, we suggest the standard use of the cultural caveat in clinical and research settings, as this may offer important cultural contexts in PGD to add to the growing literature. Also, the development and/or adjustment of existing measurement tools are necessary to increase its global applicability and use, and to reduce diagnostic discrepancies across communities. In this way, a more valid investigation of its etiology and potential (inter)cultural differences that influence grief symptoms is feasible, which may provide important cultural factors and contexts that can be proactively used by researchers and clinicians. For example, accurate identification of overlapping and dissimilar cultural aspects in PGD could increase accuracy for and cultural sensitivity in diagnosis and interventions across the globe.

Although further research is needed across cultures and populations, PGD is now a condition recognized in international classifications with specific diagnostic criteria. Events affecting communities worldwide, including the recent COVID-19 pandemic and those related to climate change, have the potential to increase the rates of PGD on the global stage. PGD treatment capacity building worldwide is needed but will likely not be enough. The development and rolling out on a large scale of a stepped-care approach to PGD seems to be the only viable solution to address this societal challenge in the long run.

**Open peer review.** To view the open peer review materials for this article, please visit http://doi.org/10.1017/gmh.2023.28.

**Data availability statement.** N/A for a review of the literature.

**Author contribution.** All authors have contributed sufficiently to the work described in this manuscript to be included as authors. All authors have read and approved the final manuscript.

**Financial support.** C.E.H. was supported by a postdoctoral grant from the Normandy region.

**Competing interest.** E.B. reports grants from the National Institutes of Health and the US Department of Defense, licenses or royalties from Springer and Wolters Kluwyer, and consulting fees from Cerevel Therapeutics. The other authors report no competing interests.

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
