## [Reviewer Report]

Dear Editors, Dear Marit,

Thank you for inviting us to submit a review article for the relaunch of the Cambridge Prisms: Global Mental Health. Hereby we would like to submit the attached manuscript entitled: “Bereavement Issues and Prolonged Grief Disorder: A Global Perspective” for consideration for publication.

Our review aimed to provide a global perspective on bereavement, grief reactions, and Prolonged Grief Disorder (PGD), a new diagnosis recently introduced into DSM5-TR. 

Our findings suggested that grief reactions seem to be consistent across different cultures, although differences and variations in the expression of symptoms may exist across cultures. Given the heterogeneity in PGD prevalence rates across the globe, specific populations may be at more risk for PGD. This may possibly be due to confounding factors including population specificities (e.g., refugees, migrants and conflict survivors). We concluded that cross-cultural reliable validation and development of PGD screening and assessment may thus be critical to support future research of grief reactions and PGD, especially outside of the Westernized civilization. We suggest that more transcultural research on PGD and the use of a stepped-care approach is necessary, as it may lead to the identification of culture-bound symptom of PGD and the adaptation of current treatment protocols, making us able to improve health at the individual level and health care systems for people who suffer from PGD. 

We hereby confirm that the data of this manuscript have not been submitted or published in another journal. All authors have contributed sufficiently to the work described in this manuscript to be included as authors. All authors have read and approved the final manuscript. To the best of our knowledge, no conflict of interest exists. 

We hope that you consider the manuscript suitable for publication in your journal. 

We are looking forward to your reply.

Sincerely,

Charlotte Hilberdink & Eric Bui 

(on behalf of all authors)

Contact details:

Charlotte Elize Hilberdink, PhD

Normandie Université 

UNICAEN, INSERM - U1237

PhIND - Physiopathology and Imaging of Neurological Disorders

NEUROPRESAGE Team (Institut Blood and Brain @ Caen-Normandie)

GIP Cyceron

14000 Caen

France

E-mail: hilberdink@cyceron.fr

---

## [Reviewer Report]

The authors have prepared a well-researched and clearly written overview of relevant and timely PGD research. My main suggestions include recommendations for additional literature and citations that I believe to be foundational for the field.

Section 1.1

Here you could include the latest prevalence rates from Prigerson et al., 2021

Section 1.1.1

This section could be reduced and complemented with a clear table presenting the diagnostic definition differences.

Section 1.1.2

‘Culture is defined as a set of traditions, rituals, values, and beliefs that are shared between

members of a group of human beings.’ Please include a specific reference

In this section you could introduce the cultural norm hypothesis and how this might explain culturally variability in the expression, duration of grief responses

Chentsova-Dutton YE, Tsai JL, Gotlib IH. Further evidence for the cultural norm hypothesis: positive emotion in depressed and control European American and Asian American women. Cultur Divers Ethnic Minor Psychol. 2010 Apr;16(2):284-95. doi: 10.1037/a0017562. PMID: 20438167; PMCID: PMC2864927.

Chentsova-Dutton Y, Maercker A. Cultural Scripts of Traumatic Stress: Outline, Illustrations, and Research Opportunities. Front Psychol. 2019 Nov 15;10:2528. doi: 10.3389/fpsyg.2019.02528. PMID: 31803094; PMCID: PMC6872530.

Section 1.4

Please incorporate the findings from this recent and relevant review: Wojtkowiak J, Lind J and Smid GE (2021) Ritual in Therapy for Prolonged Grief: A Scoping Review of Ritual Elements in Evidence-Informed Grief Interventions. Front. Psychiatry 11:623835. doi: 10.3389/fpsyt.2020.623835

Section 2

Please consider including our paper below which clearly outlines several relevant hypotheses that may explain the difference prevalence rates worldwide

Stelzer, E.-M., Zhou, N., Maercker, A., O’Connor, M.-F., & Killikelly, C. (2020). Prolonged Grief Disorder and the Cultural Crisis. Frontiers in Psychology, 10. https://doi.org/10.3389/fpsyg.2019.02982

Section 3.4

You may consider adding the literature on ambiguous and refugees experiences of multiple losses and loss of homeland/culture

Renner, A., Jäckle, D., Nagl, M., Plexnies, A., Röhr, S., Löbner, M., Grochtdreis, T., Dams, J., König, H. H., Riedel-Heller, S., & Kersting, A. (2021). Traumatized Syrian Refugees with Ambiguous Loss: Predictors of Mental Distress. International Journal of Environmental Research and Public Health, 18(8). https://doi.org/10.3390/IJERPH18083865

Comtesse, H., & Rosner, R. (2019). Prolonged grief disorder among asylum seekers in Germany: the influence of losses and residence status. European Journal of Psychotraumatology, 10(1), 1591330. https://doi.org/10.1080/20008198.2019.1591330

Discussion

Please consider referring to the International Prolonged Grief Disorder scale and the cultural supplement as a tool that researchers and clinicians may use to assess PGD for the ICD-11 with global applicability.

Kokou-Kpolou, C. K. (2021). Letter to the Editor: Prolonged grief disorder, posttraumatic stress disorder, and depression following traffic accidents among bereaved Balinese family members: Prevalence, latent classes and cultural correlates. Journal of Affective Disorders, 295, 1–2. https://doi.org/10.1016/j.jad.2021.08.007

Killikelly, C., & Maercker, A. (2023). The cultural supplement: A new method for assessing culturally relevant prolonged grief disorder symptoms | PsychArchives. Clinical Psychology in Europe. https://www.psycharchives.org/en/item/12f7905c-ede6-4cfc-ba86-c7c7f2a529d5

Killikelly, C., Zhou, N., Merzhvynska, M., Stelzer, E. M., Dotschung, T., Rohner, S., Sun, L. H., & Maercker, A. (2020). Development of the international prolonged grief disorder scale for the ICD-11: Measurement of core symptoms and culture items adapted for chinese and german-speaking samples. Journal of Affective Disorders, 277, 568–576. https://doi.org/10.1016/j.jad.2020.08.057

Please clarify the reference to anxiety here? ‘The identification of key culture bound symptoms of anxiety have previously supported their inclusion in DSM-5 (Hinton et al., 2010), and the development of specific treatment protocols (Hinton et al., 2005),’

Please consider revising your statements on the impact of COVID-19 on PGD, see

Lenferink, L. I. M., & Boelen, P. A. (2023). DSM-5-TR prolonged grief disorder levels after natural, COVID-19, and unnatural loss during the COVID-19 pandemic. Journal of Affective Disorders Reports, 12, 100516. https://doi.org/10.1016/J.JADR.2023.100516

In terms of the future directions and implications it would be interesting if the authors could propose a framework or ideas on how to foster, develop and facilitate transnational, cross-cultural research on PGD. What ideas for future research directions and initiatives might be possible and feasible?

---

## [Reviewer Report]

Thank you for giving me the opportunity to read this interesting piece of work. Both the aspect of PGD and cultural influences are of high importance. I made several suggestions which could in my opinion strengthen the manuscript. Most of them are rather minor. What I believe requires further attention is the discussion which is not as informative as it could be.

1. Abstract. It seems a bit contradictory that the sentence “Grief reactions are generally consistent across different cultures.” is followed by "differences and variation in their

expression may exist across cultures...", reformulating this passage could add to clarity.

2. The headline “From Acute Grief to Integrated Grief to Prolonged Grief” suggests that integrated grief is a necessary precondition of prolonged grief, I think this is neither the case nor intended by the authors.

3. Since the focus of this article is explicitly a global one with implications for beyond specific psychiatric settings, it is probably preferable to use terms such as mental disorders/mental health conditions instead of psychiatric conditions as recommended by both the WHO and the DSM.

4. I found it not entirely logical that after the authors introduce PGD in ICD-11 first, detailed criteria are then described for the DSM-5-TR, but not for ICD. Maybe adding a table showing all criteria for both classificatory systems and discussing differences would suffice.

5. How can ICD and DSM show “only moderate overlap” if they only “slightly differ”. Are the authors referring to overlap in diagnostic criteria or overlap in cases?

6. When talking about cultural differences, it might be helpful to differentiate expressions from symptoms (i.e. the pathological expression). At times (e.g. in paragraph 1.1) these two aspects get mixed or are used interchangeably.

7. It could makes sense to move the paragraph of prevalences (and differences in prevalences) (paragraph 3) to the beginning of the manuscript (maybe following paragraph 1.1.1) because most of the discussion of cultural differences is used to explain differences in prevalence.

8. Since PGD has only recently been introduced in the classificatory systems, it has not been part of the major international epidemiological initiatives, such as World Mental Health Surveys or Burden of Disease Studies. Also, many assessments rely on questionnaire data instead of diagnostic interviews. I somewhat missed a critical assessment of the current state of available data when presenting epidemiological data for PGD.

9. I found it somewhat difficult to understand the direct conclusions from the findings that have been presented. The authors mentioned various aspects from cultural differences in normative grief reactions to differences in risk factors. I believe the discussion could be sharpened by explaining in greater detail what follows from all these findings. Moreover, the discussion of cultural differences mainly focuses on the difference between Western and Non-western cultures, which is only a very broad distinction of potential cultural differences. Does the available data allow for a more fine-grained distinction of cultural backgrounds? And what are the main dimensions of cultural differences that are relevant for PGD? And what can we learn from these differences about GPD? Or is it just important to acknowledge them in measurement and treatment?

---

## [Reviewer Report]

The authors have been very responsive to the reviewers' comments and provided a thoughtful revision. I have nothing to add.